# Pharmacovigilance and Adverse Drug Reaction Reporting among the General Public in Lithuania: A Cross-Sectional Study

**DOI:** 10.3390/healthcare11081133

**Published:** 2023-04-14

**Authors:** Agne Valinciute, Rima Jurate Gerbutaviciene, Renata Paukstaitiene, Loreta Kubiliene

**Affiliations:** 1Department of Drug Technology and Social Pharmacy, Lithuanian University of Health Sciences, Sukileliu Ave. 13, LT-50166 Kaunas, Lithuania; 2Department of Physics, Mathematics and Biophysics, Lithuanian University of Health Sciences, Eiveniu Str. 4, LT-50161 Kaunas, Lithuania

**Keywords:** adverse drug reaction, consumers, reporting, pharmacovigilance, knowledge, safety

## Abstract

**Background**: Despite the fact that for over ten years, Lithuanian consumers have been able to report adverse drug reactions (ADR) directly to the competent authority, reporting rates remain low. A comprehensive understanding of consumer perceptions and experiences regarding ADRs is needed to ascertain further factors impacting their engagement in ADR reporting. This study aimed to assess consumer knowledge of, attitude toward, and practice of reporting ADRs. **Methods**: A questionnaire-guided cross-sectional survey among 404 consumers between October 2021 and June 2022 was conducted. The semi-structured questionnaire comprised open-ended and closed-ended questions to explore the sociodemographic characteristics and general knowledge of ADRs and pharmacovigilance. Other question items evaluated attitudes toward ADR reporting and ADR reporting practice. The data were summarised using descriptive statistics, while the chi-square test was used to assess categorical variables at *p* < 0.05. The overall percent score in the knowledge and attitude domains was divided into groups of “poor”, “moderate”, and “good” knowledge, as well as “positive” and “negative” attitudes. **Results**: While having a generally poor understanding, this study demonstrates that Lithuanian consumers have a favourable attitude toward pharmacovigilance, particularly regarding issues involving the requirement for reporting. The data also revealed the justifications for reporting and not reporting ADRs. **Conclusions**: The current study provided the first understanding of consumer awareness and ADR reporting intentions, which can help to develop educational campaigns and interventions addressing pharmacovigilance and ADR reporting.

## 1. Introduction

The method of pharmacovigilance that is most frequently utilized is spontaneous ADR reporting [1,2]. It is both a crucial element and a potent instrument of any nation’s pharmacovigilance system [3]. Spontaneous reporting is defined as: “an unsolicited communication by a healthcare professional or consumer to a company, regulatory authority or other organization (e.g., WHO, Regional Center, Poison Control Center) that describes one or more adverse drug reactions in a patient who was given one or more medicinal products and that does not derive from a study or any organized data collection scheme”. It involves various stakeholders: competent authorities, marketing authorization holders, healthcare professionals and patients [4].

At first, reporting was restricted to physicians, since it was believed that only a doctor could offer high-quality information on ADRs and minimize the risk of reporting known or unrelated associations [5,6]. Later, additional healthcare professionals (HCPs) were added to the spontaneous reporting system (SRS), including pharmacists and nurses, who today significantly contribute to the pharmacovigilance system [7,8,9]. In recent years, patients have also been permitted to report directly to the authorities, for instance, within European Union (EU) nations [10]. Pharmacovigilance underwent a significant legal shift in the EU in 2012. One of the largest changes brought about by this legislation was the requirement that patient reporting of ADRs be implemented in all EU Member States. With this action, the EU recognized patients as important sources of information regarding the safety of medicines and prepared the way for a quicker—and more thorough—collection of adverse drug reactions [11].

The spontaneous ADR reporting system in Lithuania is overseen by the State Medicines Control Agency (SMCA), which is a division of the Ministry of Health of the Republic of Lithuania. It is not only pharmacists, marketing authorization holders, and healthcare professionals who can submit ADRs directly to the SMCA. The SMCA website has had an online form that patients can use to report ADRs directly since 2013. The form can be either filled out and submitted, or it can be downloaded, filled out and delivered by email. It can also be completed by phone on a designated number [12,13]. Unless more details are needed, no individualized feedback is supplied to the reporter. Twenty patients took advantage of the opportunity to report ADRs to the SMCA during the first year after being authorized to direct report. From 2013 to 2020, there were shifts and fluctuations in the volume of patient reports, but they did not differ noticeably and stayed largely modest.

The number of ADR reports received does not accurately reflect the number of adverse drug reactions, according to one study, especially in light of the fact that more drugs were used in Lithuania during the same time period [12]. Examining notifications of ADRs received over a three-year period, excluding notifications of COVID-19 vaccines, there was a clear downward trend in the number of overall notifications of ADRs, i.e., 1739 total notifications in 2019, 1338 notifications in 2020, and 801 notifications in 2021 [14]. It is important to note that in 2021, there was a sharp rise in the number of notifications from patients and healthcare providers about using COVID-19 disease vaccines in active immunization. Additionally, the SMCA actively encouraged and promoted the reporting of all adverse responses to medicines and vaccinations at numerous meetings held throughout the year and in articles posted on significant national portals and the social media site Facebook [14].

Patients frequently have an awareness of their medical issues and medications. ADRs are nevertheless underreported, due to a lack of knowledge, a lack of clarity, and challenges with the forms and procedures for reporting ADRs [15]. Finding out what influences the number of received reports is crucial for improving patient-driven voluntary ADR reporting. Our goal was to evaluate the patients’ awareness of ADRs and their knowledge of what to do in the event that one materialized. To the best of our knowledge, this is the first quantitative study to explore and understand the knowledge, attitude, and practices regarding pharmacovigilance and ADR reporting among the general public in Lithuania.

## 2. Materials and Methods

### 2.1. Ethical Approval

The study (reference no. BE-2-59) was approved by the Kaunas Regional Biomedical Research Ethics Committee, Lithuania.

### 2.2. Study Design and Sampling

A cross-sectional study was conducted to assess the public knowledge of, attitude toward, and experience with reporting adverse drug reactions in Lithuania. Initially, the questionnaire was planned to be distributed in pharmacies to the medicine consumers who were dispensed prescribed medicines or who purchased over-the-counter medicines and who agreed to participate in the study. As a result of the current COVID-19 pandemic, the distribution of the questionnaire was too slow. To accelerate data collection, the research protocol was updated: the places in which the questionnaire was distributed were expanded to healthcare facilities and public libraries. The online tool, an anonymous questionnaire in Microsoft Forms, was also enabled to collect data by sending invitations to complete the questionnaire to e-mail addresses freely available on the Internet. The authors collected data for eight months (from October 2021 to June 2022).

A literature review was conducted before designing the questionnaire [6,16,17,18]. The questionnaire was completed by adding questions derived from qualitative research [19]. The questions and topics from the literature that were considered important were either modified or directly included as items in the questionnaire.

The pilot questionnaire was provided to volunteer consumers (*n* = 20) to assess the translation quality and the questionnaire’s understanding. Minor modifications were made to some of the questions after the pilot study. Based on the pilot study, it was estimated that the questionnaire would take 8–20 min to complete. The data collected for the pretest were included in the final analysis.

The questionnaire contained seven main sections. The first section collected the sociodemographic characteristics of the consumers, such as age, sex, education, and marital status. The second section included five multiple-choice questions designed to measure knowledge about PV and ADR reporting. A knowledge score was prepared as a guiding tool to assess the knowledge, with one point for the correct answer and zero for the wrong answer. The sum of all items gave a maximum score of 5. Consumers were categorized based on their overall knowledge scores using Bloom’s original cutoff points as “good knowledge” if the score was 80–100% (5–4 points), “moderate knowledge” if the score was 60–79% (3 points), and “poor knowledge” if the score was <60% (<3 points) of the maximum score [20]. The third section comprised 23 questions to evaluate the consumers’ attitude toward pharmacovigilance and ADR reporting. Respondents were asked to indicate their level of agreement or disagreement on a five-point Likert scale consisting of “strongly agree”, “agree”, “neutral”, “disagree”, and “strongly disagree” on the scale, valued 5 to 1, respectively. The sum of all items gave a maximum score of 95. The overall level of attitude was categorized using Bloom’s original cutoff points as a “positive attitude” if the score was 80–100% (76–95 points), “moderate attitude” if the score was 60–79% (57–75 points), and “negative attitude” if the score was less than 60% (<57 points) [20]. The fourth section contained three practice-related questions with multiple-choice options. The fifth and seventh sections included one question each that asked the consumers to rank the reasons for not reporting and the reasons for ADR occurrences, respectively. The sixth section asked the consumers to identify their information sources for ADRs.

### 2.3. Statistical Analysis

The data were edited, cleaned, coded, entered, and summarized using the statistical software IBM SPSS Statistics 29. The age did not meet the conditions of normal distribution and was therefore compared in three groups using the nonparametric Kruskal–Wallis test with pairwise comparisons, and the Mann–Whitney U-test was used to compare the age of two groups. The outcomes were presented as medians and quartiles (Q1–Q3). The relationship between the qualitative variables was analysed using the χ2 test. To assess the internal consistency reliability and construct validity of the Likert scale questionnaire on attitudes related toward pharmacovigilance and ADRs, Cronbach’s α coefficient and exploratory factor analysis were used. The Cronbach’s α coefficient ≥ 0.7 is an acceptable cutoff for internal consistency. The frequency and relative frequency were used to characterize the findings (percentage). Statistical significance was declared at *p*-value < 0.05.

## 3. Results

### 3.1. Demographic Characteristics of Participants

Four hundred and four participants (18–86 years of age) were included in the study. About 75.5% (*n* = 305) of the participants were female, and 24.5% (*n* = 99) were male, 36.6% (*n* = 148) were single, divorced or widowed, 63.4% (*n* = 256) had a partner. Respondents were divided into two groups according to their educational background: those with a university degree (77.5%, *n* = 313) and those with less than a university degree (22.5%, *n* = 91), i.e., a college, vocational, secondary or another type of education. The results of the demographic distribution of the participants are presented in Table 1.

### 3.2. Consumers’ Knowledge and Attitudes

There were five questions assessing the knowledge regarding pharmacovigilance and ADR. As shown in Table 2, 32.7% and 36.1% knew the terms pharmacovigilance and ADR, respectively. Among respondents, 71.0% correctly indicated the institution responsible for collecting ADR reports, 63.6% were aware of their possibility of reporting ADRs directly, and only 14.6% did not know the method of ADR reporting.

Consumers were categorized based on their overall knowledge scores using Bloom’s cutoff points as “good knowledge” if the score was 80–100% (4–5 points), “moderate knowledge” if the score was 60–79% (3 points), and “poor knowledge” if the score was <60% (1–2 points) of the maximum score. The results are shown in Table 2.

There were 19 questions assessing the attitudes toward ADR reporting. The overall level of attitude was categorized using a modified Bloom’s cutoff point as a “negative” attitude if the score was ≤70% (≤66.5 points) and a “positive” attitude if the score was >70% (>66.5 points). Slightly more than half (55.5%) had a “positive” attitude toward ADR reporting compared with a “negative” attitude (44.5%). The results are shown in Table 3.

An association was found between the consumers’ age groups, education, pharmacovigilance knowledge, and attitude towards ADR reporting. Consumers with the age median 30–41.50 were significantly more knowledgeable of ADRs and pharmacovigilance than other age groups [X^2^ = 20.673, *p* = 0.003]. There was a significant association between consumers with “good” knowledge and higher education and above [X^2^ = 8.727, *p* = 0.013]. Similar results were observed for attitude and knowledge. The consumers with higher education and above significantly showed more positive attitudes compared with consumers with lower education [X^2^ = 6.703, *p* = 0.035] (Table 4).

### 3.3. Consumers’ ADR Reporting Practice

The respondents were asked about their ADR reporting practice. The consumers who experienced ADR were asked additional questions. The ADR-experiencing consumers were divided into reporters and nonreporters, and the sociodemographic characteristics of both groups are shown in Table 5.

There was no statistically significant association between sex, marital status, education, and usage of medication in the last three months between the reporters and nonreporters. However, a statistically significant association was found between reporters and nonreporters in terms of the age median (Q1–Q3), which showed that older consumers who experienced ADR were more likely to report ADR (Table 5).

All consumers who experienced an ADR but who did not report it (nonreporters, *n* = 42) were questioned about the reasons for nonreporting; they stated that “ADR was not severe (40.48%), “ADR was possible” (26.19%), and “I didn’t realize it was ADR” (26.19%) (Figure 1).

The consumers who experienced an ADR and reported it (*n* = 43) were asked about the recipient of the information (Figure 2). Most of the consumers informed GPs and physicians about their ADR. It was less common to report to nurses and pharmacists.

## 4. Discussion

Direct consumer ADR reporting has been available in Lithuania since 2013 [12]. Until then, the obligation to report ADRs applied to doctors, pharmacists, and marketing authorization holders. The present study was a natural extension of an earlier qualitative study and aimed to evaluate consumers’ knowledge of, attitudes toward, and practice of pharmacovigilance and ADR reporting by applying a quantitative approach [19].

According to the results of the completed questionnaires, more than half of the consumers had poor knowledge of pharmacovigilance and ADRs. This finding confirmed one of the conclusions of the previous focus group discussions [19]. The overview of each knowledge question might give the impression that the overall score could be higher than the final results. This implication could be read as proof that consumers have heard some information about pharmacovigilance, ADRs, and reporting. However, at the same time, it also shows that information gaps are present. It is worth noting that the term “pharmacovigilance” had never been heard, and the question was answered incorrectly by around two-thirds of respondents. The qualitative study showed the same results, as none of the participants were familiar with this term [19].

Direct consumer reporting of ADR to SMCA has been possible since 2013, and until 2020, there were fewer than 350 consumer ADR reports in total [15]. In 2021, there was a spike in consumer ADR reporting (17 times more than in 2020, 76.5% of total ADR reports), associated with the COVID-19 vaccine due to the pandemic; however, consumer reporting fell to 46.5% of total ADR reports [15]. These ADR reporting results show that consumers generally know the reporting procedure if it concerns a particularly widely escalated and relevant topic. The reporting statistics support our hypothesis about knowledge gaps. It also signals that education and highlighting the importance of other groups of medicines not often discussed in media could increase the number of ADR reports.

After analysing the results, age and education level appeared to significantly affect the knowledge of the pharmacovigilance system in Lithuania and ADRs. Similar results were observed in a study performed in Portugal [21]. The mentioned study also indicated sex as a factor influencing knowledge, which was not the case in our study. These findings did, however, generally indicate that older consumers with lower levels of education had lower levels of understanding in relation to pharmacovigilance and ADRs. Many international studies have yielded comparable findings demonstrating that the general population has insufficient knowledge about pharmacovigilance [17,22,23,24,25,26,27].

Interesting results were found comparing knowledge and ADR reporting practice. There were no significant relations between sex, education level, recent medication usage, and reporting practice. Still, the age median suggested that older consumers tended to report more often than slightly younger consumers. The older population visit doctors more often than the younger population, which could be one reason that reporters’ age median was higher. In 2021, there were 441.71, 683.21, and 815.9 visits per 100 inhabitants to doctors of people aged 18–44, 45–64, and 65+, respectively [28]. Most of this study’s reporters reported the ADR to a general practitioner or physician, which also supports the hypothesis that the more contact with an HCP might lead to a report of ADR.

Most respondents reported the information to medical staff when ADRs occurred; only one reported it to a pharmacist. The tendency to report ADR to an HCP was consistent with the results of other studies [29,30,31]. On the other hand, in those studies, the consumers were more likely to report ADRs not only to their doctors, but also to pharmacists. In 2022, the competent authority in Lithuania received only four reports from pharmacists [32]. It was shown that patients tended to report ADRs to physicians because they require confirmation or recognition of the adverse event. Based on our qualitative study, consumers in Lithuania associate physicians with health conditions, whereas pharmacists are an information source about medicines in general [19,33].

It appears that consumers were motivated to report ADR for several reasons. The most important were worry about the ADR in a personal context and altruistic reasons, in order to prevent possible harm to other consumers. These results were in alignment with studies on consumers in the Netherlands, Croatia, and the UK [19,34]. The motives indicated for not reporting the ADR have also frequently been observed in other studies. The lack of severity (40.48%) and plausibility (26.19%) of the reactions has previously been reported in other studies [21,34,35]. The third reason for not reporting in the present study was a lack of knowledge and confidence in ADR recognition and discussing an ADR with a competent person. The consumers recognized a shift in their health condition, but they either stopped taking the medicine or continued without further action; therefore leaving the ADRs unreported [36,37]. Interestingly, consumers felt confident enough to report adverse reactions to the widely debated COVID-19 vaccines. This shows that trust is built on knowledge and support, both from the authorities in charge and from health staff.

We found that our survey respondents had a positive attitude toward ADR monitoring, with the proportion of consumers who had an overall positive attitude exceeding 70%. Other studies have also reached this result [25,37,38,39]. A study conducted in Korea revealed that positive attitudes toward motives, including the expectations, necessity, and duty for spontaneous reporting were associated with the intent to report ADRs [37]. Even though the attitude among Lithuanian consumers was positive, few respondents had direct reporting experience. A statistically significantly positive association was found between attitude and knowledge, where knowledge also positively acted on patient confidence to report ADRs. According to the published literature, multilayer initiatives have been introduced to enhance consumer outreach and increase the knowledge of ADR reporting behaviours in several countries [40,41]. Media attention to the advantages and ADRs of particular medications, initiatives using various social media (for example, Facebook and Instagram), and new forms of communication methods have been shown to increase consumer knowledge of reporting systems and ADRs [40,42,43,44,45,46]. Not only is empowering consumers with the knowledge and interventions needed to achieve successful outcomes, but equipping them with user-friendly and more accessible tools is essential. Developing more flexible reporting approaches and systems could help increase ADR patient reports. Since November 2015, ADR reporting in Poland has been possible through a smartphone application called Mobit Skaner. The user-friendly application supports the instant reception of all basic information about a medicine. Simultaneously, it is simple to submit ADR reports [27,47]. Additional information, such as feedback on how many similar cases have been reported, whether the reaction is known and listed in the medicine data sheet, or where to find more reliable information, could partly take some of the workloads off doctors.

To the best of our knowledge, this is the first nationwide quantitative study to explore consumers’ knowledge of, attitudes toward, and practice of ADR reporting in Lithuania. Many studies have focused on the reporting of HCPs’ and pharmacists’ perspectives on voluntary ADR reporting [48,49,50], but we believe there to be a relatively small number of studies that provide an understanding of consumers’ perspectives. It has been shown that there are differences across countries regarding factors influencing the decision to report an ADR [51]. Therefore, a study like this could add valuable insights regarding consumers’ attitudes and intentions toward spontaneously reporting ADRs and help to adapt specific interventions.

### Strengths and Limitations of This Study

The findings of this study can lead to improvements to the current state of pharmacovigilance education among consumers. Despite the valuable information obtained from this study, the following limitations need to be acknowledged. The possible limitations of this study were that the cross-sectional design of the study with only a snapshot of participants may have led to the possibility of selection bias, and the sample size was small. The participation of a more significant number of people with greater consumer diversity would increase this study’s value. Although these limits influence the generalizability of results, this study provides a knowledge base and valuable insights to improve the understanding of the pharmacovigilance system and, most importantly, the practice of ADR reporting among consumers.

## 5. Conclusions

This study shows that consumers in Lithuania, despite having mainly poor knowledge, have a positive attitude towards pharmacovigilance, especially concerning the questions related to the need for reporting. We also showed that patients who visit their doctors more frequently than other age groups were more likely to report ADRs. The results also identified the reasons for reporting and not reporting the ADRs. The present study gave the first insight into consumers’ knowledge and reporting of ADR intention, which can be useful in preparing educational campaigns and interventions regarding pharmacovigilance and ADR reporting.

## Figures and Tables

**Figure 1 healthcare-11-01133-f001:**
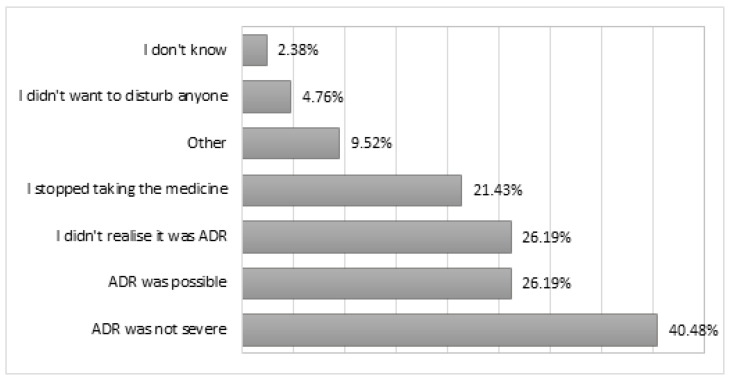
Reasons and actions taken by consumers who experienced ADR. Multiple answers.

**Figure 2 healthcare-11-01133-f002:**
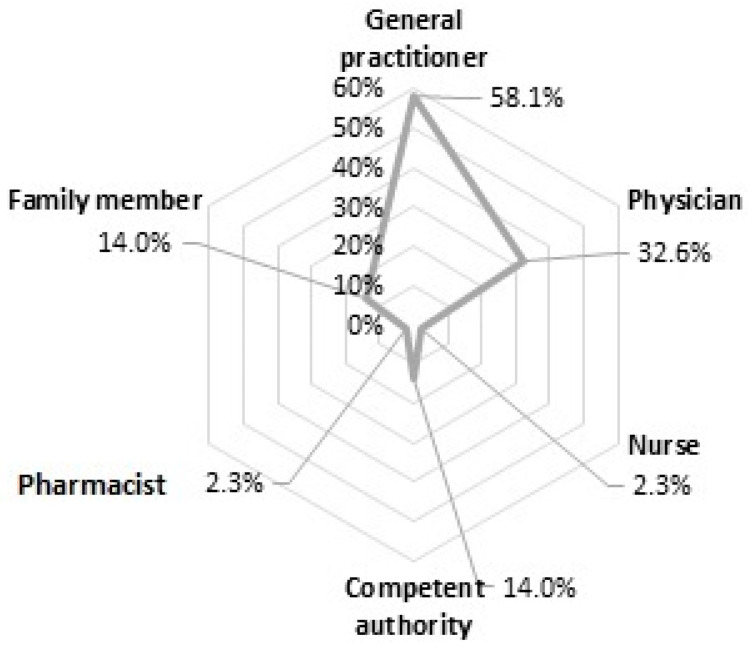
Recipients of ADR information. Multiple answers.

**Table 1 healthcare-11-01133-t001:** Demographic characteristics and clinical background of the survey respondents.

Characteristics	Frequency (*n* = 404) (%)
Age group (years)	
mean (SD)	41.77 (13.4)
min	18
max	86
median	38
≤35	141 (34.9)
35–55	186 (46.0)
>55	77 (19.1)
Sex	
Male	305 (75.5)
Female	99 (24.5)
Marital status	
Single/Divorced/Widowed	148 (36.6)
Married/Have a partner	256 (63.4)
Level of education	
Higher education and above	313 (77.5)
Advanced vocational education and training and below	91 (22.5)
Have you taken any prescription or over-the-counter medication in the last three months?	
Yes	328 (81.2)
No	76 (18.8)
Experience of the symptoms that could be described as ADR (in the past 12 months)	
Yes	57 (14.1)
No	317 (78.5)
Undecided	29 (7.2)
No answer	1 (0.2)
Experience of spontaneous ADR reporting (if experienced ADR or undecided)	
Yes	43 (50.6)
No	42 (49.4)
No answer	1 (1.2)

ADR—Adverse drug reaction; SD—Standard deviation.

**Table 2 healthcare-11-01133-t002:** Patients’ knowledge of pharmacovigilance and adverse drug reaction reporting.

Variable	Frequency, *n* (%)
Pharmacovigilance is (*n* = 404)	
A methodology for the identification, assessment, and prevention of an adverse reaction to the drug ^a^	132 (32.7)
A report about an adverse reaction	65 (16.1)
The science of improving the safety of a medicinal product	4 (1.0)
The science of evaluating the benefits and risks of a medicine	34 (8.4)
I don’t know; I’ve never come across this term	169 (41.8)
An adverse reaction to a drug is (*n* = 404)	
An unwanted negative response of the human body to a medicinal product that has been used in a normal dose for the prevention, diagnosis, or treatment of a disease or to change the physiological functions of a person, including reactions due to wrong use and noncompliance with the conditions of use of approvals, including abuse ^a^	146 (36.1)
An unwanted negative response of the human body to a medicinal product that has been used in a normal dose for prevention, diagnosis, or treatment of a disease or to change human physiological functions	145 (35.9)
All possible negative reactions of the human body to the medicinal product that are listed in the information sheet of the medicine	91 (22.5)
Name of disease/illness	4 (1.0)
None of the given options is suitable	6 (1.5)
I don’t know what that means	12 (3.0)
An adverse reaction to a medicine may be reported by ^b^ (*n* = 404)	
Healthcare professionals ^a^	318 (78.7)
Pharmaceutical professionals ^a^	54 (13.4)
Marketing Authorisation holder ^a^	187 (46.3)
Patients ^a^	257 (63.6)
None of the above is applicable	2 (0.5)
Don’t know/no opinion	19 (4.7)
Methods of adverse drug reaction reporting ^b^ (*n* = 404)	
Phone ^a^	273 (67.6)
Online form ^a^	293 (72.5)
Email ^a^	267 (66.1)
Post	94 (23.3)
Fax	63 (15.6)
Don’t know/no opinion	59 (14.6)
ADR reports are collected by (*n* = 404)	
State Medicines Control Agency ^a^	287 (71.0)
Ministry of Health of The Republic of Lithuania	40 (9.9)
Center for Infectious Diseases and AIDS	1 (0.2)
Institute of Hygiene	1 (0.2)
None of the given options is suitable	13 (3.2)
I don’t know/I have no opinion	62 (15.3)
Sources for obtaining information about adverse drug reaction (*n* = 403) ^b^	
Journals	18 (4.5)
Internet	208 (51.5)
Physician	208 (51.5)
Pharmacy Specialist	162 (40.2)
Medicine leaflet	351 (87.1)
Knowledge evaluation (*n* = 404)	
Good	51 (12.62)
Moderate	87 (21.53)
Poor	266 (65.84)

ADR—Adverse drug reaction; SD—Standard deviation. ^a^ correct answer, ^b^ multiple responses.

**Table 3 healthcare-11-01133-t003:** Attitudes of patients’ towards pharmacovigilance and adverse drug reaction reporting.

Statements	Strongly Disagree, *n* (%)	Disagree, *n* (%)	Neutral, *n* (%)	Agree, *n* (%)	Strongly Agree, *n* (%)
I don’t think it makes sense to report an adverse reaction if it is known	102 (25.2)	129 (31.9)	60 (14.9)	86 (21.3)	27 (6.7)
I believe that one report of an adverse reaction has no impact	106 (26.2)	170 (42.1)	67 (16.6)	45 (11.1)	16 (4)
I believe that reporting an adverse reaction prevents me from having more serious health problems *	15 (3.7)	53 (13.1)	82 (20.3)	176 (43.6)	78 (19.3)
I believe that by reporting an adverse reaction, I can protect others *	10 (2.5)	10 (2.5)	36 (8.9)	193 (47.8)	154 (38.1)
All serious adverse reactions are known before the medicine is marketed	35 (8.7)	128 (31.7)	125 (30.9)	95 (23.5)	21 (5.2)
I must always inform my doctor and pharmacist of any adverse reactions I experience *	11 (2.7)	21 (5.2)	49 (12.1)	199 (49.3)	124 (30.7)
I lack knowledge about possible adverse reactions to my medicines	30 (7.4)	140 (34.7)	109 (27)	99 (24.5)	23 (5.7)
Before reporting an adverse reaction, it is necessary to make sure that it is related to the medicine	5 (1.2)	28 (6.9)	79 (19.6)	210 (52)	81 (20)
I think it is a waste of time to inform my doctor or pharmacist about adverse reactions to my medicine	125 (30.9)	204 (50.5)	49 (12.1)	17 (4.2)	9 (2.2)
The leaflet is a useful resource for information on adverse reactions *	8 (2)	7 (1.7)	9 (2.2)	202 (50.1)	177 (43.9)
If I inform my doctor or pharmacist about an adverse reaction, it is likely that my treatment will be changed *	18 (4.5)	15 (3.7)	85 (21)	216 (53.5)	70 (17.3)
I am afraid that I may face legal consequences if I report an adverse reaction incorrectly	122 (30.2)	166 (41.1)	95 (23.5)	13 (3.2)	8 (2)
I find it difficult to talk to doctors or pharmacists about adverse drug reactions I have experienced	96 (23.8)	184 (45.5)	75 (18.6)	42 (10.4)	7 (1.7)
My doctor does not take my complaints about possible adverse reactions seriously	72 (17.9)	155 (38.5)	131 (32.5)	36 (8.9)	9 (2.2)
When prescribing the medicine, the doctor tells me about all the advantages and disadvantages of the treatment, possible risks, and side effects *	37 (9.2)	123 (30.4)	85 (21)	135 (33.4)	24 (5.9)
I do not report an adverse reaction because I am not sure if it is an adverse reaction	25 (6.2)	109 (27.1)	141 (35.1)	113 (28.1)	14 (3.5)
Admitting to the doctor that the medication prescribed by her/him caused the adverse reaction would reduce my confidence in her/his professionalism	81 (20.1)	178 (44.2)	100 (24.8)	36 (8.9)	8 (2)
The pharmacist usually provides a hurried service without much interest	35 (8.7)	151 (37.6)	109 (27.1)	89 (22.1)	18 (4.5)
It is the patient’s responsibility to report an adverse reaction to a medicine *	14 (3.5)	57 (14.2)	85 (21.2)	186 (46.4)	59 (14.7)

* These variables were automatically reverse coded to improve reliability. Cronbach’s alpha coefficient of 0.72 indicates acceptable reliability. Attitude (*n* = 391)—positive >70% (>66.5 points), *n* = 217 (55.5%); negative ≤70% (≤66.5 points), *n* = 174 (44.5%).

**Table 4 healthcare-11-01133-t004:** Descriptive and statistical interference related to age, education, and attitude.

Category	Knowledge Count, *n* (%)			*p*-Value
	Poor	Moderate	Good	
Education				
Higher education and above	196 (73.1)	74 (85.1)	43 (87.8)	0.013
Advanced vocational education and training and below	72 (26.9)	13 (14.9)	6 (12.2)	
Attitude				0.035
Negative (≤70%)	78 (72.9)	23 (21.5)	6 (5.6)	
Positive (>70%)	180 (63.4)	61 (21.5)	43 (15.1)	
Age	39 (33.25–53.75) *	37 (30–52)	35 (30–41.50) *	0.003

* pairwise comparisons *p*-value 0.006.

**Table 5 healthcare-11-01133-t005:** Demographic characteristics of the consumers who experienced ADR.

Characteristics		ADR Reporting Practice		*p*-Value
		Nonreporters, *n* (%)	Reporters, *n* (%)	
Sex	Female	32 (76.2)	34 (79.1)	0.750
	Male	10 (23.8)	9 (20.9)	
Marital status	Single/Divorced/Widowed	17 (40.5)	14 (32.6)	0.448
	Married/Have a partner	25 (59.5)	29 (67.4)	
Education	Higher education and above	11 (26.2)	10 (23.3)	0.754
	Advanced vocational education and training and below	31 (73.8)	33 (76.7)	
Usage of medication in the last three months	Not used	4 (9.5)	2 (4.7)	0.381
	Used	38 (90.5)	41 (95.3)	
Age	Median (Q1–Q3)	37.5 (32.75–51.5)	48 (35–62)	0.049

## Data Availability

The data presented in this study are available on request from the corresponding author. The data are not publicly available due to confidentiality policies.

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
