# Peer review of "Pharmacovigilance and Adverse Drug Reaction Reporting among the General Public in Lithuania: A Cross-Sectional Study"

_healthcare, 2023, doi:10.3390/healthcare11081133_

Round 1

Reviewer 1 Report

The manuscript is very well written, based on good research. Only minor changes are needed, as reported in appendix.

Author Response

Thank you for the review. Please find the answered comments and updated manuscript.

Reviewer 2 Report

Reviewer opinion
The manuscript is well written of interest to provide and develop pharmacovigilance (PV) reports from the general population. It positions itself into the perspective of PV at the European continent scale to better follow public health potential drugs iatrogenic reactions. Therefore the manuscript fully deserves to be published and should open to further investigations and publications to support Lithuanian national endeavour for PV public reporting in addition to HCP reporting.

Reviewer suggestions of improvement
Line 36
 : please replace « ADR reporting. » by « ADR reporting »
Line 42: please replace “patients     – input” by “patients - input“
Line 68 to 74: please correct the following sentences as they repeat 2 time the same things. The authors should remove the 2nd sentence as showed herafter “Examining notifications of ADRs 68 received over a three-year period, excluding notifications of COVID-19 vaccines, there was a clear downward trend in the number of notifications of ADRs, i.e., 1,739 total noti fications in 2019, 1,338 notifications in 2020, and 801 notifications in 2021 [15]. However, the overall number of notifications of ADRs received throughout the three-year period— 1739 in 2019, 1338 in 2020, and 801 in 2021—showed a definite decreasing trend in the number of notifications of NDVs. This excluded notifications of COVID-19 vaccinations [15].
Line 157 : (optionnal) The reviewer would suggest to better explain (in a short sentence) the two differencial levels of education

Author Response

PleaseResponse to Reviewer 2 Comments
Reviewer opinion
The manuscript is well written of interest to provide and develop pharmacovigilance (PV)
reports from the general population. It positions itself into the perspective of PV at the
European continent scale to better follow public health potential drugs iatrogenic reactions.
Therefore the manuscript fully deserves to be published and should open to further
investigations and publications to support Lithuanian national endeavour for PV public
reporting in addition to HCP reporting.
Response: Many thanks for your kind words!
Reviewer suggestions of improvement
Line 36 : please replace « ADR reporting. » by « ADR reporting »
Line 42: please replace “patients – input” by “patients - input“
Line 68 to 74: please correct the following sentences as they repeat 2 time the same things.
The authors should remove the 2nd sentence as showed herafter “Examining notifications of
ADRs 68 received over a three-year period, excluding notifications of COVID-19 vaccines,
there was a clear downward trend in the number of notifications of ADRs, i.e., 1,739 total
noti fications in 2019, 1,338 notifications in 2020, and 801 notifications in 2021 [15].
However, the overall number of notifications of ADRs received throughout the three-year
period— 1739 in 2019, 1338 in 2020, and 801 in 2021—showed a definite decreasing trend
in the number of notifications of NDVs. This excluded notifications of COVID-19
vaccinations [15].”
Line 157 : (optionnal) The reviewer would suggest to better explain (in a short sentence) the
two differencial levels of education
Response: Thank you for your review. We agree with the suggestions, and they were
implemented in the manuscript. see the attachment.

Reviewer 3 Report

The manuscript “Pharmacovigilance and Adverse Drug Reaction reporting  among the general public in Lithuania: A Cross-sectional Study“ is in the scope of the Journal „Healthcare“, but some minor issues should be addressed before publication:

1.       Line 74 – what is NDV?

2.       In the Introduction part or in the Discuisson part there are no data whether some education through mass media or social media regarding the reports of ADR was conducted in Lithuania about the reporting ADRs by patients. Has the Ministry of Health undertaken some steps to inform the patients about the forms for ADRs report: how to report, how to fill the form, what to report, what data are necessary to put in the report etc…

3.       Table 1

Under question “Experience of the symptoms that could be described as ADR (in the past 12 months)“ there are  57 (14,1) Yes and  29 (7,2) Undecided which is total of 86 (57+29) volunteers, while under question „Experience of spontaneous ADR reporting (if experienced ADR or undecided)„ there are Yes 43 (50,6) and  No 42 (49,4) which is total of 85 (43+42) volunteers with experienced or undecided ADR. So if in the previous question 86 had ADRs or undecided, in the latest it seems that you have lost one patient or the sum should be again 86 instead 85. There some inconsistences.

4.       This part of the conclusion, lines 311-317

„To the best of our knowledge, this is the first nationwide quantitative study that explored consumers' knowledge of, attitude toward, and practice of ADR reporting in Lithuania. Many studies have focused on reporting HCPs' and pharmacists' perspectives on voluntary ADR reporting [53-55]; we believe that a relatively small number of studies  provided an understanding of the consumers' perspectives. It was shown that there were  differences across countries in factors for the decision to report an ADR [56]. Therefore, a  study like this could add valuable insights into consumers' attitudes and intentions toward spontaneously reporting ADRs and help adapt specific interventions“

belongs to the discussion part – you can not put in a conclusion a line „It was shown that there were  differences across countries in factors for the decision to report an ADR [56].“ making a conclusion about practice in other countries if in your study you have note conducted research in other countries. Conclusion of a study should answer the aim of your study and should refer to your results. Comparison with the results of other studies belong to discussion part.  

The other part of the conclusion is acceptable although more specific sentences addressing the aim of the study may improve the conclusion

Round 2

Reviewer 3 Report

The paper contains all the required changes and has been improved after being revised.